# OpenReview forum: "Secret Alignment: Reframing Backdooring as Security Primitive in the Personal AI Era"
_ICLR.cc/2026/Conference — Submitted to ICLR 2026_

### Official Review · Reviewer_jYqP · 2025-10-31

**Soundness:** 2
**Presentation:** 1
**Contribution:** 1
**Rating:** 0
**Confidence:** 4

**Summary:**

This paper introduces Secret Alignment, a lens through which backdoor deployments are viewed not as attacks, but rather as ways to encode specific behaviors in models in a positive way. The authors analyze three backdoor approaches—SafeTrigger, Instructional Fingerprinting, and SudoLM—through the six key properties of Secret Alignment, which are, in order, effectiveness, which asks whether the trigger reliably induces the intended behavior; harmlessness, which asks whether the model's performance is negatively affected by the presence of the backdoor; persistence, which asks whether the backdoor is robust to weight updates in the model; efficiency, which asks the computational cost of making the backdoor; robustness, which asks whether the backdoor can withstand adversarial inputs that aren't the expected trigger; and reliability, which asks about unintended risks. The authors then discuss the "behavioral structure" and "decision complexity" of Secret Alignment.

**Strengths:**

The paper explains relevant background material well. The replications of existing papers are very valuable, especially as the paper reports different results (as in section 4.1)!

**Weaknesses:**

I think this paper falls somewhat short of its promise. For one, it's not actually clear to me why the positive reframing of backdoors matters; many papers in the backdoor literature already discuss effectiveness, robustness, reliability, etc. The concept of "Secret Alignment" is introduced a lot, but the paper does not, I believe, justify why introducing this concept is necessary. Perhaps it would be useful as a way of taxonomizing different backdoor approaches, but the paper only considers three backdoors. Finally, it seems like Draguns et al. (2024) perfectly satisfies 4.1, 4.2, 4.4 and 4.5 by construction, and 4.3 empirically, so it seems hard to argue that existing literature falls short of the described criteria.


Minor comments:

- It would be nice to parenthesize the citations (e.g. ~\citep)
- Line 12: I would use "and" instead of "or" here
- The use of the word "compromise" as a synonym of "attack" is a somewhat awkward
- In Figure 1, the caption says "Instructionak". Also, I would put the labels directly on the plot, so that the reader doesn't need to cross-reference the numbers
- Figure 1 does not explain what the levels 1–5 are
- The definition of backdoors seems quite restrictive, and is limited to only "string prepending" (lines 180–181).

**Questions:**

- On line 44, you say "the security concerns (aligning with human values)", but the examples are "model thefts, unauthorized accesses, and malicious misuses"; these seem unrelated to human value alignment, or am I missing something?
- Can you elaborate on the threat model described on lines 61–64?

---

### Official Review · Reviewer_nk67 · 2025-10-31

**Soundness:** 3
**Presentation:** 3
**Contribution:** 2
**Rating:** 4
**Confidence:** 4

**Summary:**

This paper introduces a unifying framework for backdooring techniques repurposed for security, focusing on settings where private LLMs are deployed. The authors systematically evaluate three representative methods: SudoLM (access control), Instructional Fingerprinting (ownership verification), and SafeTrigger (defending against finetuning attacks) across six properties: effectiveness, harmlessness, persistence, efficiency, robustness, and reliability. Through extensive empirical evaluation, they reveal brittleness in all three methods, and reveal some discrepancies from the original claims. The paper proposes a behavioral taxonomy based on behavior density (sparse vs clustered) and decision complexity (simple vs multi-level), which helps explain empirical failures and suggests that sparse and simple associations perform best.

**Strengths:**

- The framework provides a unifying lens for analyzing disparate backdooring methods. The behavioral taxonomy (density and complexity) offers predictive insights for future designs.
- Extensive empirical work with comprehensive ablation studies, e.g. pruning ratios, training loss effects, optimization dynamics. Multiple model architectures were tested for generalization.
- Generally well-written, with figures effectively communicating key findings.
- Addresses an important emerging class of positive backdooring techniques, providing actionable guidance for practitioners considering such methods.

**Weaknesses:**

1. **Replication validity**: SudoLM results show substantial discrepancies from the original paper (Fig 2), but no explanation is provided for why. This raises questions about implementation correctness or training adequacy that could undermine confidence in the findings.
2. **Motivation for threat models**: Lines 47-51 describe cryptographic alternatives as "costly and complex" with insufficient justification. What specific cryptographic methods were considered? Given that access control (SudoLM) assumes black-box query access, why not simply gate API access with standard authentication? The threat model needs stronger justification for when backdooring is necessary vs conventional access control.
3. **Novelty**: The three methods and six evaluation properties are from prior work. While systematic evaluation has value, the core contribution is primarily empirical replication + taxonomy, which may be insufficient for ICLR.
4. **Related work**: The findings could benefit from additional context by extending the related work, with papers broadly in the positive backdooring category e.g. “Improving Alignment and Robustness with Circuit Breakers”. Additionally, given the framing of a dichotomy of cryptography vs backdoors (Lines 047-051, 054-055, 1591), the existence of cryptographic LLM backdoors (e.g. Goldwasser 2022, Draguns 2024) seems potentially relevant to some of the analysis and recommendations.

**Minor Comments:**
- Line 327: "Objective Overhead" undefined
- “NeurIPS” is listed in the references but does not appear cited in text
- Reference “Martins (2024)”: in the references author appears as “Daniel Martins” but the linked site lists the author as “Pedro Martins”.
- Terminology: "Harmlessness" is defined as preserving performance, conflicting with papers that use the term to mean not producing harmful outputs. "Reliability" somewhat confusingly refers to unintended risks. Standard terminology would improve clarity, even if the current meaning is consistent with the paper introducing the properties and renaming would require using more than one word per property.
- Formatting issues: e.g. typos on Lines 117, 183, 194, 206; layout issues, e.g. Fig 7 caption

**Questions:**

1. Fig 2 shows SudoLM performing much worse than reported. What might have caused the discrepancy with the original papers? What differs between your setup and the original?
2. How were Fig 1 radar ratings determined? Is this subjective assessment or based on quantitative thresholds?
3. Line 180: Why is it assumed that triggers are always prepended?
4. Line 459: Are you claiming that adding a new fingerprint necessarily overrides/ignores existing ones, or can they coexist?

---

### Official Review · Reviewer_VhjG · 2025-11-01

**Soundness:** 2
**Presentation:** 3
**Contribution:** 2
**Rating:** 2
**Confidence:** 3

**Summary:**

This paper examines positive backdooring in the context of the Personal AI paradigm. The authors argue that this paradigm introduces new security challenges that traditional centralized defenses cannot adequately address. To analyze how backdoors can be used constructively rather than maliciously, they introduced Secret Alignment to conceptualize protective backdoors as a covert mapping between secret triggers and intended model behaviors. Through three representative use cases, the authors empirically show that current positive backdooring methods suffer from brittleness of the trigger-behavior relationship. They further identify two key behavioral factors to explain the shortcomings.

**Strengths:**

1. The paper conducts a unified replication of three representative positive backdoor techniques under consistent model architectures and experimental settings. It improves fairness and reproducibility across prior works.
2. The paper provides a careful empirical evaluation of prior claims, uncovering overstatements and identifying behavioral trade-offs (e.g., degradation of general performance), contributing to a more realistic understanding of proactive backdoor.

**Weaknesses:**

1. My main concern is that the technical depth of the paper is relatively limited. The proposed “Secret Alignment” framework provides a conceptual reinterpretation of backdoor, but it does not introduce new algorithmic mechanisms or theoretical derivations. The formalization of trigger-based behavior $f(s+q) → r_2$ is also standard in prior backdoor literature. The six evaluation properties are more descriptive rather than technically grounded. Thus, the contribution is limited to the conceptual and empirical level rather than methodological.
2. While “Secret Alignment” is a neat way to tie things together, the current body of work for positive backdoor seems too small (only three papers cited) and scattered for this to feel fully justified. The framework comes across as something built after the fact rather than emerging naturally from prior research, so the motivation needs to be further justified to show the necessity of the work.
3. While empirical evaluations are solid, the paper lacks a deeper mechanistic understanding or modeling of why brittleness occurs (e.g., gradient dynamics, representational drift), limiting technical depth.

**Questions:**

1. Could you clarify the motivation of the proposed framework to frame positive backdooring, especially given that only a few existing works explore this area? For example, are there real-world use cases or deployed systems that currently rely on proactive backdooring for model protection in Personal AI? How realistic is the assumption that Personal AI systems will rely on such mechanisms for protection, compared to more conventional security defenses, e.g., watermaking or encryption?
3. Beyond reporting performance drops and activation inconsistencies, did you analyze internal representations including embedding shifts, neuron activation patterns to understand why certain triggers fail or interfere with normal behaviors?
4. How does Secret Alignment compare with other protective mechanisms like watermarking, model fingerprinting, or cryptographic license enforcement? Why should backdoor-based methods be preferred?

---

### Official Review · Reviewer_9mxj · 2025-11-01

**Soundness:** 3
**Presentation:** 3
**Contribution:** 2
**Rating:** 4
**Confidence:** 3

**Summary:**

This paper investigates the use of backdooring techniques for security purposes in the emerging "Personal AI" era, where individuals and small organizations own and deploy their own LLMs.

The authors reframe these techniques under a new unifying framework called "Secret Alignment," which they define as the practice of embedding covert trigger-behavior mappings into a model to enable legitimate security functions like access control, ownership verification, and safety enforcement.

Through a systematic evaluation of three representative methods (SuboLM, Instructional Fingerprinting, and SafeTrigger) across six key properties, the paper reveals that while promising, these methods are significantly brittle in practice, struggling with stability, robustness, and reliability.

**Strengths:**

1. The paper offers a novel perspective by proposing the unified "Secret Alignment" framework, which systematizes various protective backdoor techniques. It shifts the viewpoint from "backdoor as an attack vector" to "backdoor as an alignment technique" .
2. This study pioneers a six-dimensional evaluation framework, breaking the field's long-standing limitation of relying on single performance metrics .
3. The experimental design in Chapter 4 demonstrates novelty and provides substantial data support .
4. In robustness assessment, the authors innovatively developed a progressive attack scheme: by constructing six progressively similar test prompt levels, they verified that the Instructional Fingerprinting method exhibits over 50% false activation rate under approximate trigger conditions .

**Weaknesses:**

1. All experiments use only the Llama2-7B model, so the generalizability of current conclusions needs further validation.
2. The experimental threat models are oversimplified and fail to reflect real-world complexity. SudoLM evaluation lacks testing for advanced attacks (e.g., trigger reverse-engineering via API analysis); SafeTrigger testing assumes white-box attacker knowledge but ignores realistic black-box scenarios where attackers only control training data.
3. For Instructional Fingerprinting, the paper notes forgery risks but does not properly test multiple fingerprint injection. These limitations lead to overly optimistic robustness assessments that underestimate real adversarial capabilities.
4. The paper has limited originality: its main contribution is systematically replicating and evaluating existing methods (SudoLM, IF, SafeTrigger). This work is essentially rigorous benchmarking rather than groundbreaking research—more an "evaluation report" than an exploratory study—lacking substantive theoretical or methodological innovation.

**Questions:**

1. Could authors provide comparative results of the six core properties (effectiveness, harmlessness, etc.) across different model architectures/parameter scales to verify the generalizability of the conclusions?
2. The core contribution of this study focuses on the systematic replication and evaluation of existing methods (SudoLM, IF, SafeTrigger). Under the Secret Alignment framework, are there any unmentioned theoretical breakthroughs or methodological optimizations?
3. Fig. 3 shows that the Instructional Fingerprinting method has a false activation rate exceeding 50% under trigger similarity Levels 3-6, but the paper does not deeply explain the reasons. For example, whether the model over-learns template features rather than the trigger itself?
4. The paper classifies Secret Alignment into "simple association", "single-level classification", and "multi-level classification" based on "decision complexity". What are the quantitative bases for the classification criteria?

---

### Note · Authors · 2026-01-26

I have read and agree with the venue's withdrawal policy on behalf of myself and my co-authors.

---

> ### Note · Program_Chairs · 2026-01-26
>
> We approve the reversion of withdrawn submission.

---

### Meta-Review · Area_Chair_eunr · 2026-01-02

**Summary:**

The paper examines backdooring as a positive security tool rather an adversarial attack in the context of the Personal AI paradigm. The concept of Secret Alignment is introduced and experiments are conducted to evaluate three backdoor approaches.

Main limitations of the paper:
1. Limited originality. The main contribution is systematically replicating and evaluating existing methods.
2. Limited Experiments. Use only the Llama2-7B model; the threat models are oversimplified and fail to reflect real-world complexity; results for previous work show substantial discrepancies from the original paper (Fig 2), but no explanation is provided;
3. The paper proposed that there exist new security challenges in the Personal AI era, but theses challenges were not specifically described, comparing to the general security problem.

**Reviewer Concerns:**

The following concerns are essential and cannot be addressed satisfiability in a short period of time.
1. Limited originality. The main contribution is systematically replicating and evaluating existing methods.
2. Limited Experiments. Use only the Llama2-7B model; the threat models are oversimplified and fail to reflect real-world complexity; results for previous work show substantial discrepancies from the original paper (Fig 2), but no explanation is provided;

**Reviewer Scores:**

Reviewer VhjG's score is little bit low and can be increased.
Reviewer jYqP's score is too low and I believe that the reviewer did not read the detail of the paper.

---

### Decision · Program_Chairs · 2026-01-26

Reject